# Innovation and Evaluations of 3D Printing Resins Modified with Zirconia Nanoparticles and Silver Nanoparticle-Immobilized Halloysite Nanotubes for Dental Restoration

**Karwan Rashid Darbandi [1,\*] and Bassam Karem Amin [2,\*]**

1   General Directorate of Health, Ministry of Health, Kurdistan Regional Government, Erbil 44001, Iraq
2   Department of Conservative Dentistry, College of Dentistry, Hawler Medical University, Erbil 44001, Iraq
\*   Correspondence: karwan.rashid@hmu.edu.krd (K.R.D.); bassam.amin@hmu.edu.krd (B.K.A.)

**Abstract:** Additive manufacturing technologies can be used to fabricate 3D-printed dental restorations. In this study, we evaluated the effectiveness of the functionalized loading of zirconium dioxide ($ZrO_2$) nanoparticles and silver-nanoparticles-immobilized halloysite (HNC/Ag) nanotubes into 3D printing resins. We created 3D printing resins by adding different mass fractions of $ZrO_2$ and HNC/Ag. First, six groups of samples containing $ZrO_2$ were prepared, comprising five groups with different mass fractions and one control group of $ZrO_2$ containing 1 to 16 %wt. Different mass fractions of HNC/Ag fillers were combined with the $ZrO_2$ mixture and resin at the ideal ratio from 1 to 7.5 %wt. The mechanical characteristics of 3D resin that we assessed were the flexural strength, flexural modulus, fracture toughness, and the microhardness. Additional rates of $ZrO_2$ 4 %wt. and HNC/Ag 5 %wt. significantly increase the flexural strength, flexural modulus, and fracture toughness compared to the control group ($p < 0.001$). $ZrO_2$ 16 %wt. and HNC/Ag 5 %wt. were found to be significantly harder compared to the other groups ($p < 0.001$). The amounts of NPs that can be added to 3D printing resin modification appears to be 4 %wt., and HNC/Ag 5 %wt. can be advantageous in terms of fracture toughness, flexural strength, and flexural modulus. All additions of nanoparticles raised the resin's hardness.

**Keywords:** 3D resin printing; zirconia nanoparticles; silver nanoparticles; halloysite nanotube; 3D printing dental restoration





## 1. Introduction

Additive manufacturing (sometimes referred to as 3D printing) is a process that creates intricate 3D objects by layering various materials such as metals, polymers, and ceramics. This process builds the object incrementally based on a digital design file, unlike traditional subtractive manufacturing techniques. This revolutionary technology has diverse applications across multiple industries, enabling rapid prototyping, intricate geometries, and more sustainable manufacturing processes [1–4].

The advancement of 3D printing has been rapid in recent years. With its increased accuracy and durability, it is the preferred option for numerous healthcare sectors including medicine, dentistry, orthopedics, and the general creation of medical devices [5,6].

By using this technology, computerized 3D models may be effectively translated into products that are tangible. First, a digital file in standard tessellation language (STL) format needs to be created. Then, the design is printed by assembling, bonding, or polymerizing small-volume components [7,8]. Currently, there are numerous 3D printing methods in use, including vat photopolymerization (which encompasses digital light processing (DLP)) and laser stereolithography (SLA). SLA utilizes an ultraviolet (UV) laser to cure and solidify photo-polymeric resins, making it highly regarded for its precision and accuracy [9].

However, because of its superior biocompatibility, zirconium dioxide ($ZrO_2$) nanoparticles are one of the best ceramic materials used in prosthetics and dentistry [10,11]. Moreover,

$ZrO_2$ has brilliant strength and fracture toughness, two properties that are often mutually exclusive in most materials. Furthermore, findings demonstrated that various ceramic- and polymer-based composites' thermomechanical, physical, and biological qualities were improved by the inclusion of $ZrO_2$ particles [12]. Researchers in dentistry investigated the potential benefits of incorporating $ZrO_2$ particles into resin-based materials. To improve their mechanical properties they explored factors such as surface treatment, particle size, and the method of dispersion. Their findings show that adding $ZrO_2$ particles improves flexural strength and hardness [13,14].

In addition to the improvement of mechanical qualities provided by silver nanoparticles, one of the main elements influencing the therapeutic use of composite resins is antibacterial activity. Since silver nanoparticles (AgNPs) have stronger mechanical qualities and higher antibacterial activities than bare resin, they can effectively reinforce composite resins. AgNPs exhibited homogeneous dispersion in composite resins [15]. Compared to ionic silver, AgNPs have been shown to have higher antibacterial activity and less toxicity [16]. AgNPs can attain a significant antibacterial capacity with a relatively low concentration of AgNPs in the composite resin due to their high surface area-to-volume ratio [17]. The purpose of this study was to assess the mechanical properties (fracture toughness, flexural strength, modulus, and microhardness) of 3D printed resins that has been modified by different ratios of $ZrO_2$ Nanoparticles (0, 1, 2, 4, 8, and 16 %wt.) and HNC/Ag (0, 1, 2.5, 5, and 7.5 %wt.).

## 2. Materials

### 2.1. Silver Nanoparticles and Zirconia Nanoparticles

We obtained analytical-grade pure silver nitrate ($AgNO_3$), sodium borohydride ($NaBH_4$), and trisodium citrate ($C_6H_5O_7Na_3$) from (Himedia Pvt. Ltd., Mumbai, India). The zirconia nanoparticles are zirconium(IV) isopropoxide ($Zr(OiPr)_4$), propanol, and ammonium hydroxide ($NH_4OH$), and were obtained from Sigma Aldrich, Darmstadt, Germany. Halloysite was supplied by Merck with the product no. 685445F. 10-Methacryloyloxydecyl dihydrogen phosphate (MDP) and camphor quinone (CQ) were obtained from Aladdin, Shanghai, China, and 4-dimethylamino-benzoic acid ethyl ester (EDMAB) was obtained from Aladdin, Shanghai, China. Without additional purification, A1 saremco print CROWNTEC was obtained from SAREMCO Dental AG, Rebstein/Switzerland. All components were utilized as starting materials.

### 2.2. Preparation of Zirconia Nanoparticles

A meticulous approach was employed for the experimental synthesis of zirconia nanoparticles. Initially, a solution was prepared by dissolving 0.1 mM of zirconium(IV) $Zr(OiPr)_4$ in a solvent mixture consisting of 2 M propanol and 0.8 M water. Subsequently, a carefully calculated quantity of $NH_4OH$ was introduced into the solution in order to induce the formation of a dense gel comprised of $Zr(OH)_2$. The addition of $NH_4OH$ was controlled to ensure the formation of the desired structure. Following the formation of the $Zr(OH)_2$ gel, this thermal treatment served to transform the gel into zirconia nanoparticles, driven by the removal of water and organic moieties, resulting in the solid-phase formation of $ZrO_2$ nanoparticles [18]. In the process of surface-treating nano-zirconia fillers, the MDP was meticulously prepared. This treatment involved the combination of 10 %wt. of MDP with 88.8 %wt. of absolute ethanol coupled with 0.9 %wt. of EDMAB and 0.3 %wt. of CQ. Furthermore, an addition of 100 %wt. of zirconia nanoparticles was added and mixed on a stirrer for 15 min at room temperature. To ensure homogeneity and uniform dispersion of the various components, ultrasonic treatment was applied at an intensity of 150 watts for 10 min [19].

### 2.3. Preparation of Silver Nanoparticle-ImmobilizedHalloysite Nanotubes

Aqueous solutions of freshly prepared A 50 mL aqueous solution of freshly prepared 10 mM NaBH4 were introduced into a 50 mL solution containing 0.01 M $AgNO_3$ under

the presence of 100 mL of a 0.001 M sodium citrate solution. This was followed by a thorough washing regimen with deionized water. Subsequently, the dispersion was once again subjected to a controlled centrifugation procedure to ensure the purification of silver particles.

Organosilane-modified halloysite nano clays (HNCs) were then prepared. First, 25 mL of APTES was dissolved in a beaker containing toluene. Then, 2.5 g of HNCs were added to the solution. The mixture was then kept under ultra-sonication for 30 min, and the reaction solution was constantly stirred for 24 h at 70 °C. The resultant precipitated HNCs obtained after 24 h were thoroughly washed with toluene to eliminate the excess organosilane, and were then dried for 48 h at room temperature.

To immobilize silver (Ag) nanoparticles on HNC, 10 g of halloysite was dispersed in 100 mL of deionized water by a magnetic stirrer. Then, 1 g of $AgNO_3$ was introduced in the halloysite dispersed solution while it was stirred at 60 °C. Thereafter, 0.0629 g of $NaBH_4$ was added into 100 mL of deionized water. The mixture color was changed from yellow to brown/red, confirming the formation of the required nanocomposite.

The final product was collected through the filtration process and was washed three times with hot water to remove unreacted silver ions. It was dried at 80 °C for 24 h [20,21].

### 2.4. Preparation of Sample and 3D Printing

The 3D-printed resin was obtained through the subsequent incorporation of materials. We first prepared six groups of printable resin samples with varying $ZrO_2$ content, which included one control group and five groups with various mass percentages of $ZrO_2$ (Z1 (0%), Z2 (1%), Z3 (2%), Z4 (4%), Z5 (8%), and Z6 (16%)). Additionally, we created five groups with different mass fractions of HNC/Ag fillers that were mixed with the optimal composition of resin and $ZrO_2$ mixture (ZS1 (0%), ZS2 (1%), ZS3 (2.5%), ZS4 (5%), and ZS5 (7.5%)). The specific composition details of the $ZrO_2$/HNC/Ag-loaded dental printable resins are shown in Tables 1 and 2.

**Table 1.** Groups based on varying mass fractions of MDP-conditioned zirconia nanoparticles to 3D printing resin.

| Sample Code | %wt. of $ZrO_2$ | Resin |
|:---:|:---:|:---:|
| Z1 | 0 | A1 saremco print CROWNTEC |
| Z2 | 1 | A1 saremco print CROWNTEC |
| Z3 | 2 | A1 saremco print CROWNTEC |
| Z4 | 4 | A1 saremco print CROWNTEC |
| Z5 | 8 | A1 saremco print CROWNTEC |
| Z6 | 16 | A1 saremco print CROWNTEC |

**Table 2.** Groups according to different mass fractions of silver nanoparticle-immobilized halloysite nanotubes to 3D printing resin.

| Sample Code | %wt. of HNC/Ag | %wt. of $ZrO_2$ | Resin |
|:---:|:---:|:---:|:---:|
| ZS1 | 0 | 4 | A1 saremco print CROWNTEC |
| ZS2 | 1 | 4 | A1 saremco print CROWNTEC |
| ZS3 | 2.5 | 4 | A1 saremco print CROWNTEC |
| ZS4 | 5 | 4 | A1 saremco print CROWNTEC |
| ZS5 | 7.5 | 4 | A1 saremco print CROWNTEC |

The preparation procedures involved gradually adding nanoparticle proportions into the printable resin solutions whilst continuously stirring magnetically by machine for 24 h. Afterward, the solution underwent sonication for 30 min in a distal water bath. To 3D print with the same setup using a DLP printer (Phrozen Sonic Mini 4K, Hsinchu 30091, Taiwan), an LED light source with a wavelength of 385 nm is used. Following the printing process, the produced specimens were immersed in 90% isopropyl alcohol for 5 min, adhering to

the manufacturer guidelines. The specimens were carefully removed from the building platform using a scraper. To remove any last traces of uncured monomers from the surface, a second rinse using fresh isopropanol was used. Compressed air was then used to dry the specimens. The polymerization process was then aided by a post-curing treatment that lasted 10 min and used a light curing device (Solidilite V, Shofu Dental GmbH, Ratingen, Germany) with a broad wavelength spectrum spanning from 400 to 550 nm. The sample size for the study was determined based on the results of a previous study [5], resulting in a minimum sample size of 15 per group. Each group has a total of 495 specimens for the study, with 165 samples for flexural strength and modulus (n = 15), 165 samples for Vickers microhardness (n = 15), and 165 samples for fracture toughness (n = 15) [5,22]. The finished printed samples are shown in Figures 1–3.

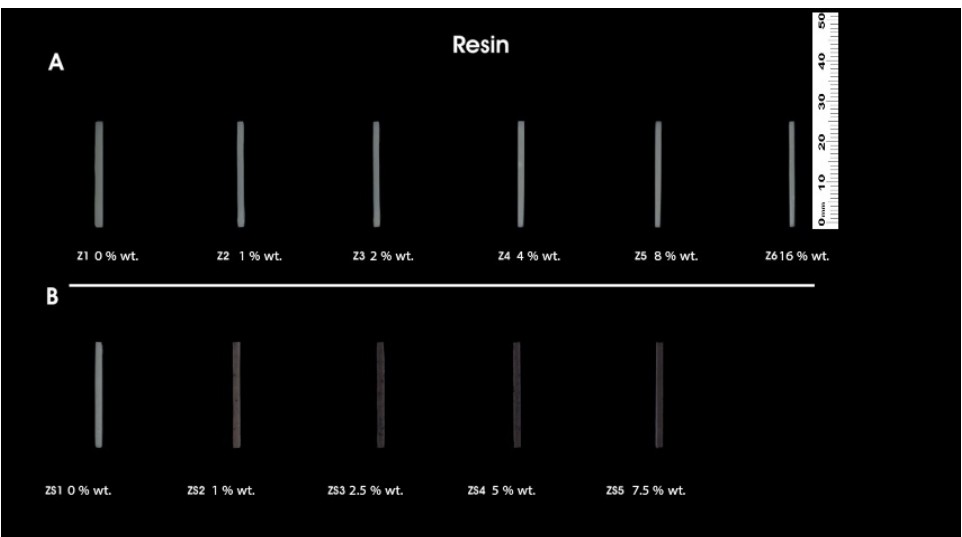

**Figure 1.** 3D-printed samples for flexural strength and flexural modules. (**A**) Unmodified 3D-printed resin (Z1); $ZrO_2$ 1 %wt. (Z2); $ZrO_2$ 2 %wt. (Z3); $ZrO_2$ 4 %wt. (Z4); $ZrO_2$ 8 %wt. (Z5); $ZrO_2$ 16 %wt. (Z6); (**B**) HNC/Ag 0 %wt. (ZS1); HNC/Ag 1 %wt. (ZS2); HNC/Ag 2.5 %wt. (ZS3); HNC/Ag 5 %wt. (ZS4); HNC/Ag 7.5 %wt. (ZS5).

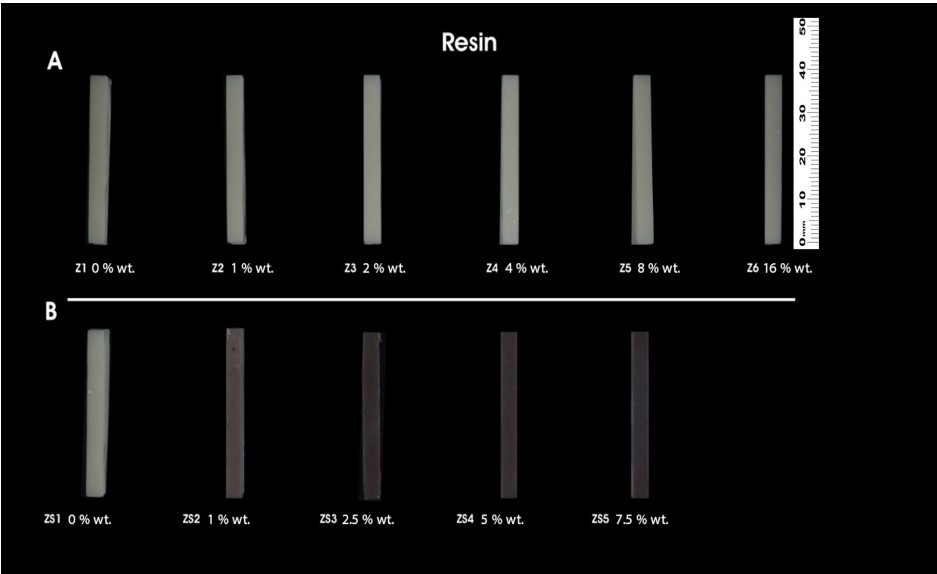

**Figure 2.** 3D-printed samples for fracture toughness. (**A**) Unmodified 3D-printed resin (Z1); $ZrO_2$ 1 %wt. (Z2); $ZrO_2$ 2 %wt. (Z3); $ZrO_2$ 4 %wt. (Z4); $ZrO_2$ 8 %wt. (Z5); $ZrO_2$ 16 %wt. (Z6); (**B**) HNC/Ag 0 %wt. (ZS1); HNC/Ag 1 %wt. (ZS2); HNC/Ag 2.5 %wt. (ZS3); HNC/Ag 5 %wt. (ZS4); HNC/Ag 7.5 %wt. (ZS5).

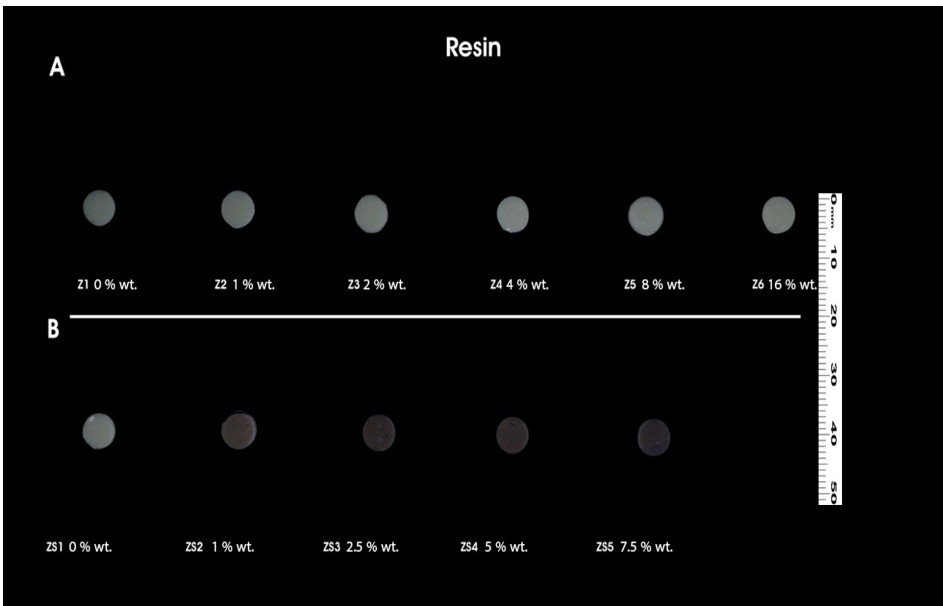

**Figure 3.** 3D-printed samples for Vickers microhardness. (**A**) Unmodified 3D-printed resin (Z1); $ZrO_2$ 1 %wt. (Z2); $ZrO_2$ 2 %wt. (Z3); $ZrO_2$ 4 %wt. (Z4); $ZrO_2$ 8 %wt. (Z5); $ZrO_2$ 16 %wt. (Z6); (**B**) HNC/Ag 0 %wt. (ZS1); HNC/Ag 1 %wt. (ZS2); HNC/Ag 2.5 %wt. (ZS3); HNC/Ag 5 %wt. (ZS4); HNC/Ag 7.5 %wt. (ZS5).

### 2.5. Flexural Strength and Modulus

Variations in the flexural strength and modulus of the 3D-printed specimens were characterized through a 3-point bend test using a universal testing machine (SANTAM; STM20 Iran, compliant with ISO standard 4049 [23]). Each group was assigned 15 specimens, with dimensions of (25 mm × 2 mm × 2 mm). After polymerization was examined for any defects, the specimens were removed from further testing and kept in distilled water at 37 °C for one full day. Each specimen was tested by mounting it on a 3-point fixture with a 20 mm support distance and applying a load at a crosshead speed of 1 mm/min until it fractured. Using the following formula, the flexural strength ($\sigma$) was computed in Mpa and flexural modulus (*E*) in GPa [24].

$$\sigma = \frac{3F1}{2bh^2}$$

$$E = \frac{F_1 l^3}{4bh^3 d}$$

where *F* is the maximum applied load (N); *l* is the distance (mm) between the supports; *b* is the width of the test specimen (mm); *h* is the height of the specimen (mm); $F_1$ is the load (N) at a point in the straight-line portion of the load/deflection curve; and *d* is the deflection (mm) at load $F_1$.

### 2.6. Fracture Toughness Testing

To assess fracture toughness, the 3D-printed resin specimens were stored in distilled water at 37 °C for 24 h. The samples were examined for defects, and those that revealed any were removed before testing and storage. Specimens were measured using the ISO 20795 standard and the identical 3-point bending fixture, as previously mentioned [24]. A total of 15 specimens were assigned per group with dimensions of 39 mm × 8 mm × 4 mm, with a 32 mm-in-length predefined notch made in the center. Subsequently, every specimen was placed onto a fixture with a support span of 32 mm and a displacement rate of 1 mm/min.

The fracture toughness ($K_I$c) was calculated using the greatest force recorded according to the following formula [24].

$$K_{Ic} = \frac{f\ \mathrm{Fmax}l}{bh^{\frac{3}{2}}} \times \sqrt{10^{-3}}$$

where $f = \frac{3x^{1/2}1.99 - x(1-x)(2.15 - 3.93x + 2.7x^2)}{\left[2(1+2x)(1-x)^{3/2}\right]}$, $x = a/h$, Fmax is the maximum recorded load (N) at fracture; $l$ is the support span distance (mm); $b$ is the width of the specimen (mm); $h$ is the height of the specimen (mm); and a is the crack length (mm).

### 2.7. Vickers Microhardness Test (VHN)

Vickers hardness number (VHN) is a measurement and report of the surface microhardness of the 3D-printed resin specimens. Prior to testing, specimens that were disk-shaped and had a diameter of 6 mm and a thickness of 2 mm were examined for any deformities. If the specimens were deformed in any way, they were eliminated. Successful specimens were kept in distilled water at 37 °C for 24 h using a 1000 gm weight (Leco Co., Michigan, MI, USA).

VHN is achieved by dividing the load applied over the surface area made by the indentation using this equation:

VHN = p/d2 × C.
VHN = Vickers microhardness number.
P = Load applied equal 1000 gm.
d2 = Diagonal length square of the indentation.
C = Constant equals 1.854.

Each disk had its top and bottom indented, and mean values were measured and statistically analyzed [25].

### 3. Result

#### 3.1. Flexural Strength (FS) and Flexural Modulus (FM)

According to the Shapiro-Wilks test, all groups met the normality assumption which was calculated and presented as a statistic test ($p$-value). This led us to utilize the parametric tests to investigate the hypothesis of this paper.

There were statistically significant variations between the various mass fractions of $ZrO_2$ and HNC/Ag fillers. Table 3 shows the standard deviations for the flexural strength and flexural modulus for each group. The findings showed that every research group successfully surpassed the minimum flexural strength requirement ($\geq$50 MPa) for crown and bridge material ISO standard 10477 [26]. The FS of the dental resins experienced a relatively negligible increase with the incorporation of 1 %wt. and 2 %wt. in comparison to the unmodified printed resins with FSs of 103.840, 104.150, and 103.190, respectively. Conversely, the presence of 4 %wt. significantly improved the FS to 128.140 and had the highest mean value. Although higher $ZrO_2$ amounts of 8 %wt. and 16 %wt. were incorporated, the FS value fell to 112.430 and 110.410.

**Table 3.** Mean and standard deviation illustration of mechanical features.

| Nanoparticles Portions | Flexural Strength (MPa) | Flexural Modulus (GPa) | Fracture Toughness (Mpa.m1.2) | Vickers Microhardness (HV0.05) |
|---|---|---|---|---|
| $ZrO_2$ | | | | |
| Z1 0 %wt. (Control) | 103.190 ± 0.769 | 4.860 ± 0.079 | 1.960 ± 0.096 | 16.090 ± 0.642 |
| Z2 1 %wt. | 103.840 ± 0.940 | 5.120 ± 0.419 | 1.970 ± 0.018 | 16.880 ± 0.576 |
| Z3 2 %wt. | 104.150 ± 1.054 | 5.360 ± 0.459 | 1.980 ± 0.049 | 17.030 ± 0.222 |
| Z4 4 %wt. | 128.140 ± 0.395 | 8.560 ± 0.562 | 2.420 ± 0.074 | 21.340 ± 0.730 |
| Z5 8 %wt. | 112.430 ± 0.217 | 7.070 ± 0.582 | 2.030 ± 0.018 | 23.440 ± 0.704 |
| Z6 16 %wt. | 110.410 ± 0.523 | 6.330 ± 0.764 | 1.920 ± 0.081 | 27.080 ± 0.391 |

**Table 3.** *Cont.*

| Nanoparticles Portions | Flexural Strength (MPa) | Flexural Modulus (GPa) | Fracture Toughness (Mpa.m1.2) | Vickers Microhardness (HV0.05) |
|---|---|---|---|---|
| ZrO$_2$/HNC/Ag | | | | |
| ZS1 0 %wt. | 128.137 ± 0.395 | 8.557 ± 0.561 | 2.422 ± 0.074 | 21.339 ± 0.730 |
| ZS2 1 %wt. | 128.540 ± 0.668 | 8.853 ± 0.177 | 2.520 ± 0.097 | 23.272 ± 0.534 |
| ZS3 2.5 %wt. | 127.778 ± 1.438 | 8.856 ± 0.450 | 2.597 ± 0.194 | 23.621 ± 0.811 |
| ZS4 5 %wt. | 132.727 ± 0.731 | 9.903 ± 0.075 | 3.001 ± 0.020 | 27.560 ± 0.714 |
| ZS5 7.5 %wt. | 121.947 ± 0.690 | 6.993 ± 0.281 | 2.796 ± 0.217 | 25.511 ± 0.924 |
| ANOVA Test | <0.001 | <0.001 | <0.001 | <0.001 |

Figure 4A shows the distribution of fillers. The 4 %wt. addition recorded the highest FS when its distribution was located at the highest value on the *x*-axis Further tests were also applied between each pair, although no difference can be seen between 1 %wt. and 2 %wt., as seen in Figure 4B.

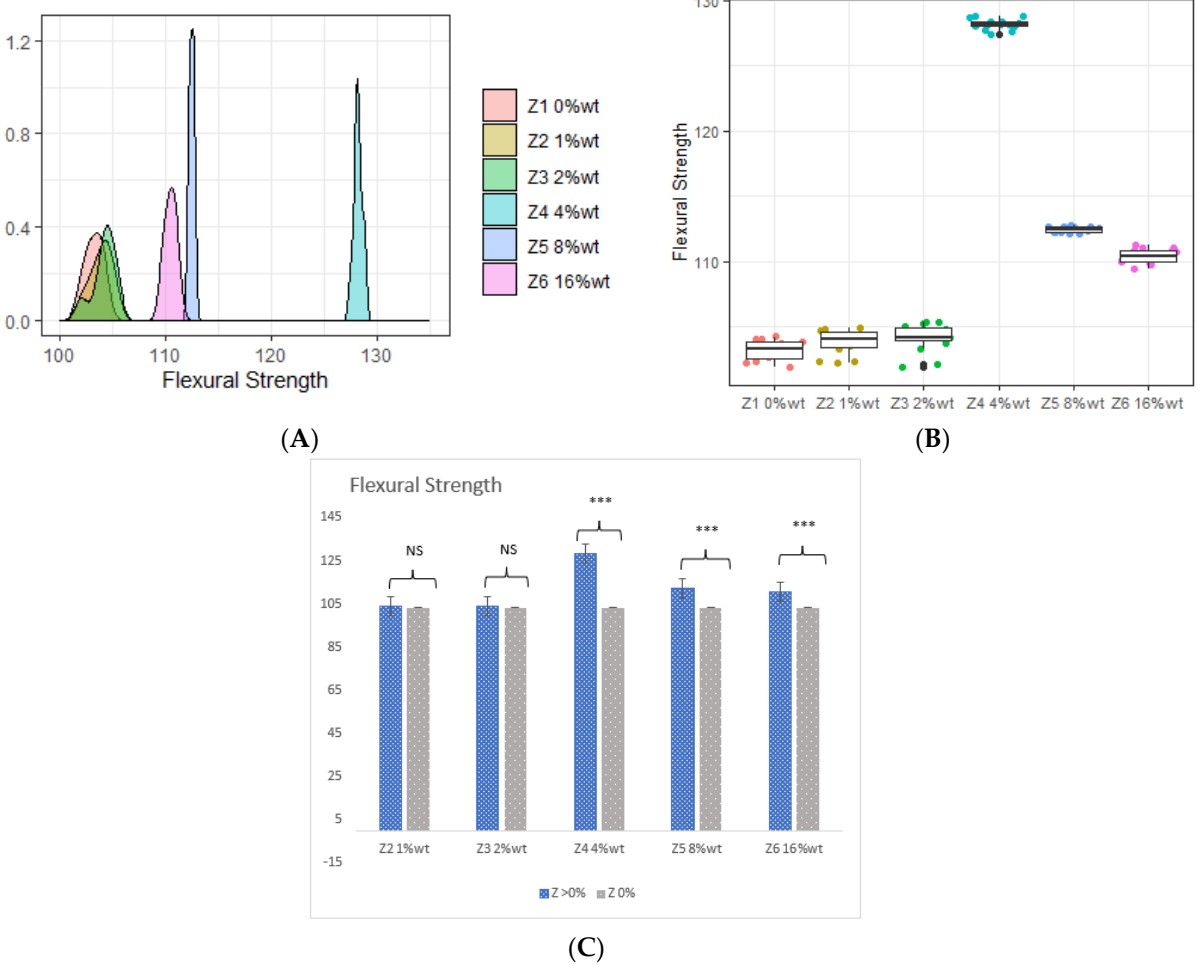

**Figure 4.** (**A**) Distribution of flexural strength in response to different proportions of ZrO$_2$ filler; (**B**) box-plot of flexural strength in response to different proportions of ZrO$_2$ filler; (**C**) displaying significant differences of unmodified resins and different proportions of ZrO$_2$ filler. *** highly significant; NS, non-significant.

The ONE-WAY ANOVA test showed significantly higher FS with 4 %wt., 8 %wt., and 16 %wt. ZrO$_2$ fractions compared to the control, as can be seen in Figure 4C.

Referring to Table 3, the FS undergoes a steady increase with the incorporation of ZS2 1 %wt. compared to the control resin, which records values of 128.137 and 128.540 by percentage change (0.3%). However, after the addition of silver up to 2.5%, the reverse occurs with the property going down by 0.3% to an overall mean value of 127.778.

Conversely, the presence of ZS4 5 %wt. showed a significant boost, reaching a peak mean FS of 132.727 where the mean value increased by 3.6%. Despite the incorporation of higher percentages of ZS5 7.5 %wt., the FS values declined by 4.8% to 121.947. Figure 5A illustrates the distribution of the fillers, with the ZS4 5 %wt. addition recording the highest FS. Its distribution is notably located at the highest *x*-axis value. Additionally, subsequent tests were conducted for each pair, revealing no significant difference between 1 %wt. and 2.5 %wt., however, a statistically significant distinction emerged among the remaining pairs, as evident in Figure 5B.

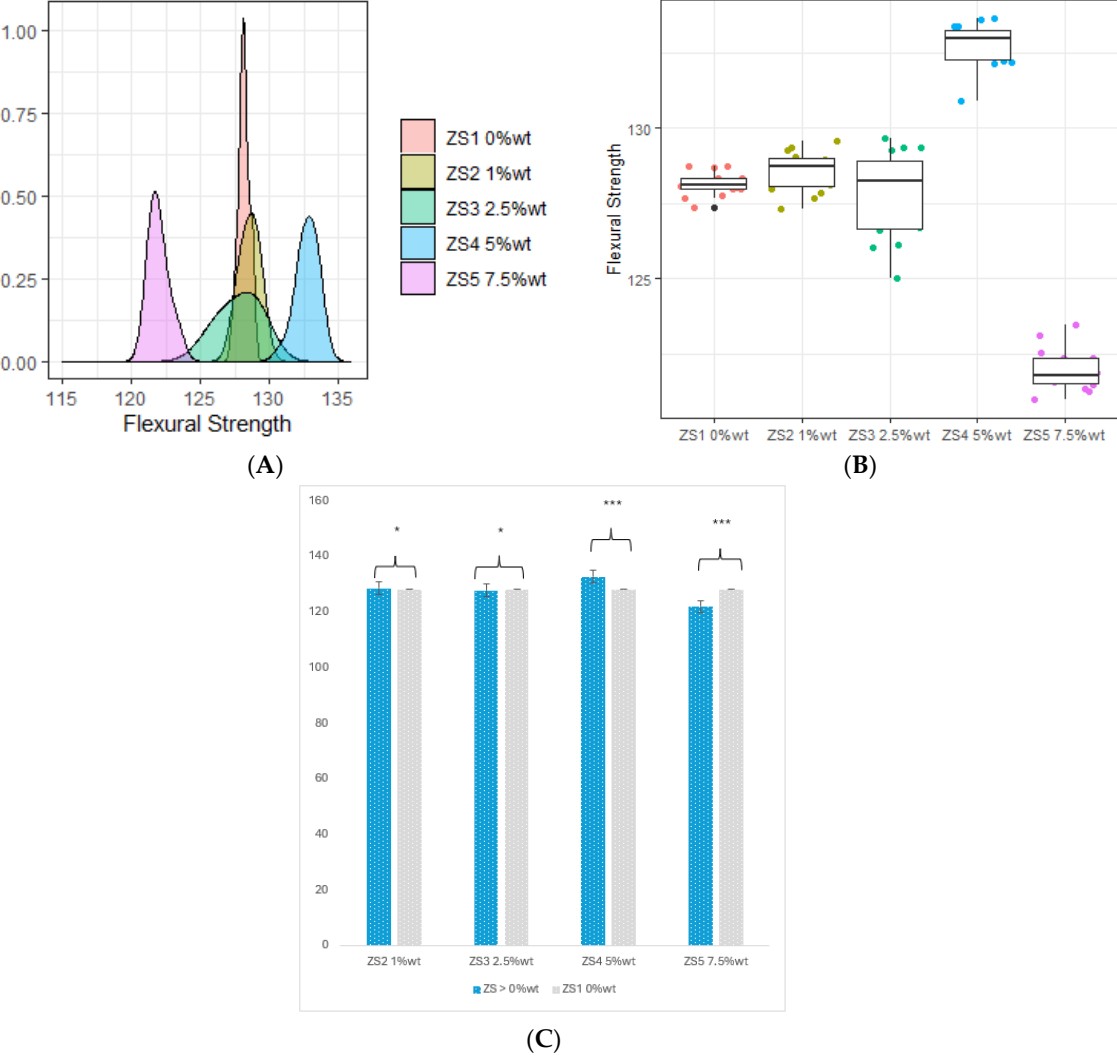

**Figure 5.** (**A**) Distribution of flexural strength in response to different proportions of silver nanoparticle filler; (**B**) box-plot of flexural strength in response to different proportions of silver nanoparticle filler; (**C**) displaying significant differences of ZrO$_2$ 4 %wt. resins and different proportions of HNC/Ag filler. * Significant difference; *** highly significant difference.

Furthermore, the ANOVA test produced a statistically significant result between the control resin and other fillers of extra ZS, indicating the impact of ZS proportions on FS. The findings emphasize a significant growth in the property with ZS fractions of ZS2 1 %wt. and ZS4 5 %wt., contrasting both fractions of ZS3 2.5 %wt. and ZS5 7.5 %wt.

### 3.2. Flexural Modulus

Table 3 shows a slight change in flexural modulus features at 1 %wt. and 2 %wt. compared to blank resin (0 %wt.), with mean values of 5.120, 5.360, and 4.860, respectively. However, 4 %wt. filler secured the highest mean value of flexural modulus (8.560), while amounts of $ZrO_2$ at 8 %wt. and 16 %wt. in specimens resulted in a decline in flexural modulus values (7.070 and 6.330, respectively). Figure 6A shows that the addition of $ZrO_2$ filler with 4 %wt. had a greater effect on flexural modulus. Furthermore, additional tests were conducted between each pair, in which significant differences were observed between all other remaining pairs as illustrated in Figure 6B. Additionally, the ANOVA test yielded a statistically significant result, indicating variations in the effects of $ZrO_2$ proportions on flexural modulus percentage usage. According to Figure 6C, blank resin illustrated no statistical differences with the $ZrO_2$ 1 %wt. filler, meaning that the flexural modulus stayed the same, whereas (compared to specimens that had additional $ZrO_2$ filler with 4 %wt., 8 %wt., and 16 %wt.) the differences ended statistically significant.

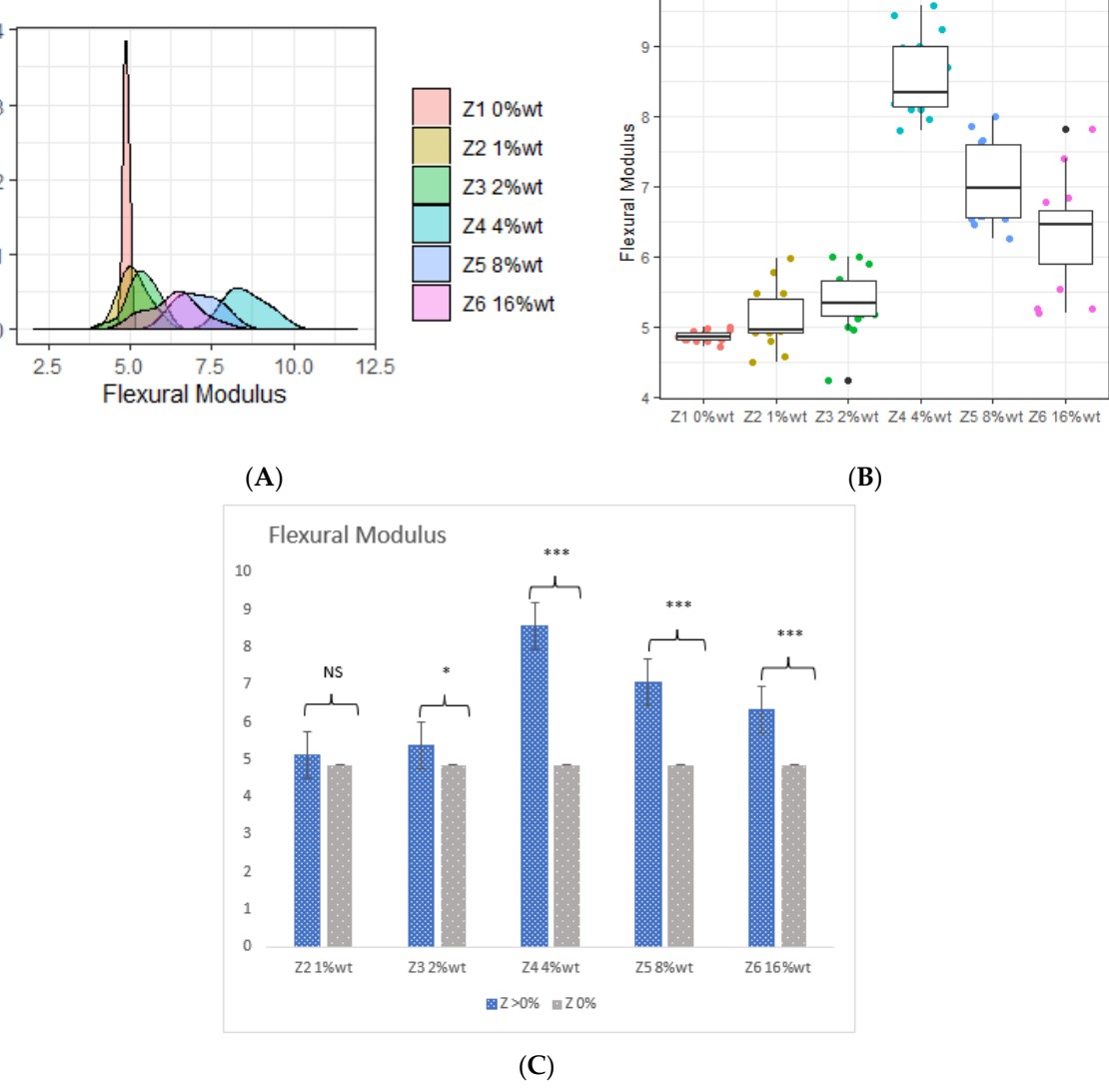

**Figure 6.** (**A**) Distribution of flexural modulus in response to different proportions of $ZrO_2$ filler; (**B**) box-plot of flexural strength in response to different proportions of $ZrO_2$ filler; (**C**) displaying significant differences of unmodified resins and different proportions of $ZrO_2$ filler for flexural modulus. * Significant difference; *** highly significant difference; NS, non-significant.

The flexural modulus of dental resins displayed a consistent upward trend upon the introduction of ZS2 1 %wt. and ZS3 2.5% compared to the control resin. The values were recorded as 8.557, 8.853, and 8.856, respectively, reflecting a marginal percentage change of 3.5%. In contrast, the inclusion of ZS4 5 %wt. demonstrated a noteworthy enhancement, attaining a peak mean FS of 9.903, signifying a substantial 15.7% increase in mean strength. Despite the integration of a higher ZS5 7.5 %wt. percentage, there was a notable decline of 18.3% in flexural modulus values, which reached 6.993. Figure 7A depicts the distribution of fillers, with the ZS4 5 %wt. addition registering the highest flexural strength. Its distribution is prominently situated at the highest *x*-axis value. Furthermore, the ANOVA test produced a statistically significant result, as illustrated in Figure 7C. The unaltered resin exhibited no statistically significant differences from the ZS2 1 %wt. and ZS3 2.5 %wt. fillers, suggesting a sustained flexural modulus. Conversely, in comparison to specimens featuring supplementary fillers at ZS4 5 %wt. and ZS5 7.5 %wt., the disparities observed were statistically significant. Furthermore, additional tests were conducted between each pair. Significant differences were observed between all other remaining pairs, as illustrated in Figure 7B.

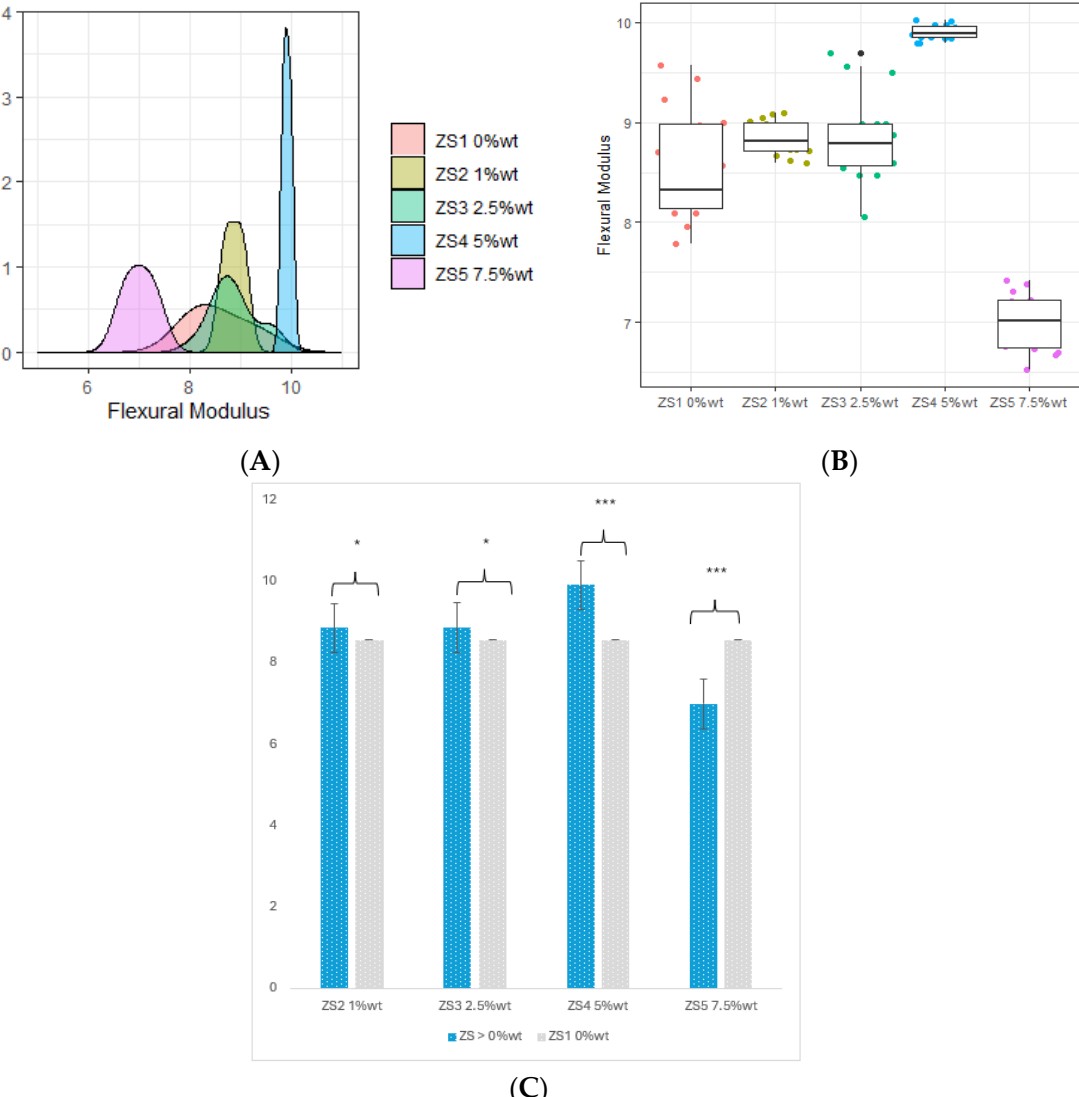

**Figure 7.** (**A**) Distribution of flexural modulus in response to different proportions of silver nanoparticle filler; (**B**) box-plot of flexural modulus in response to different proportions of silver nanoparticle filler; (**C**) displaying significant differences of $ZrO_2$ 4 %wt. resins and different proportions of silver nanoparticle filler. * Significant difference; *** highly significant difference.

### 3.3. Fracture Toughness

Compared to the unmodified specimen resin, enhanced toughness was observed in the fracture load capacity of all reinforced specimens. An upward trend can be seen between the fracture toughness and filler concentration as per Figure 8A, and a small number of changes in fracture toughness were detected after mixing 1 %wt. and 2 %wt. of $ZrO_2$ with the filler. Notably, specimens incorporating 4% filler exhibited higher fracture toughness compared to the blank resin and the 1% and 2% $ZrO_2$ additions with a mean value of 2.420 (changed by 23%). Like the other previous properties, fracture toughness was also decreased when additional $ZrO_2$ was involved in specimen resin at 8 %wt. and 16 %wt., with mean values of 2.030% as well as 1.920, respectively. Concerning the test comparison (ANOVA), a statistically significant outcome was produced only between the filler of 4 %wt. $ZrO_2$. As depicted in Figure 8C, the blank resin exhibited no statistical differences among the groups of fillers with 1 %wt., 2 %wt., 8 %wt., and 16 %wt. with unmodified resin control (0 %wt.) as seen in Figure 8B.

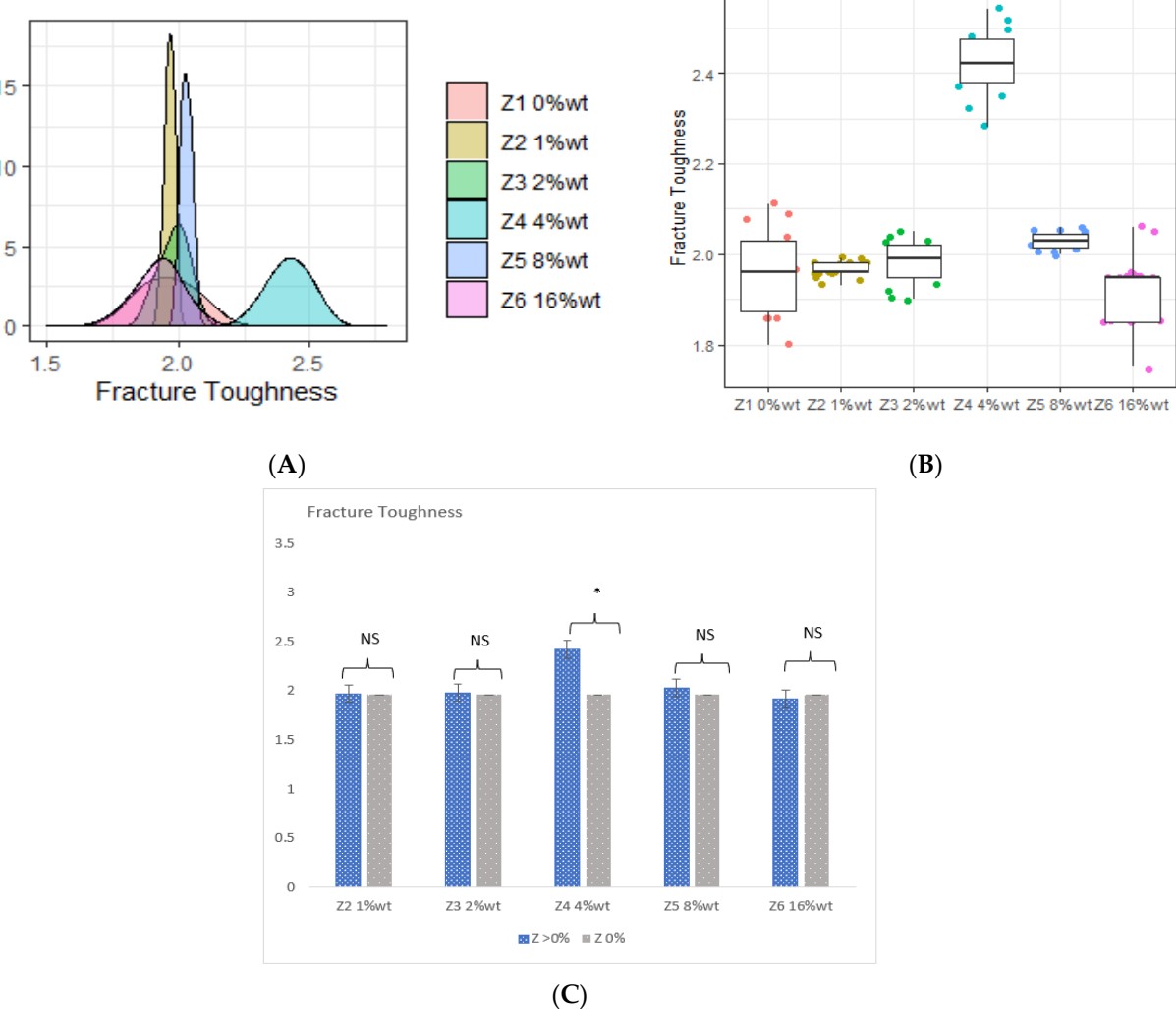

**(A)**

**(B)**

**(C)**

**Figure 8.** (**A**) Distribution of fracture strength in response to different proportions of $ZrO_2$ filler; (**B**) box-plot of fracture strength in response to different proportions of $ZrO_2$ filler; (**C**) displaying significant differences of unmodified resins and different proportions of $ZrO_2$ filler for fracture toughness. * Significant difference; NS, non-significant.

The fracture toughness of dental resins upon the introduction of ZS2 at 1 %wt. and ZS3 at 2.5%, in contrast to the control resin. The recorded values stood at 2.422, 2.520, and

2.597, respectively, indicating a marginal percentage shift of 4% and 7.2%. On the other hand, the inclusion of ZS4 at 5 %wt. showcased a remarkable advance, achieving a peak mean fracture toughness of 3.001, which is a 23.9% increase in mean strength.

Despite the infusion of a higher ZS5 percentage at 7.5 %wt., there was a noticeable dip of 15.4% in fracture toughness values, settling at 2.796. Figure 9A captures the filler distribution with the addition of ZS4 at 5 %wt. standing out with the highest fracture toughness. Its distribution is prominently positioned at the peak of the *x*-axis. Concerning the ANOVA test comparison, statistical significance was found between the control resin and the filler concentrations of silver nanoparticles except ZS2 1 %wt. filler, as illustrated in Figure 9C. There were also significant differences between all pairs, as shown in Figure 9B.

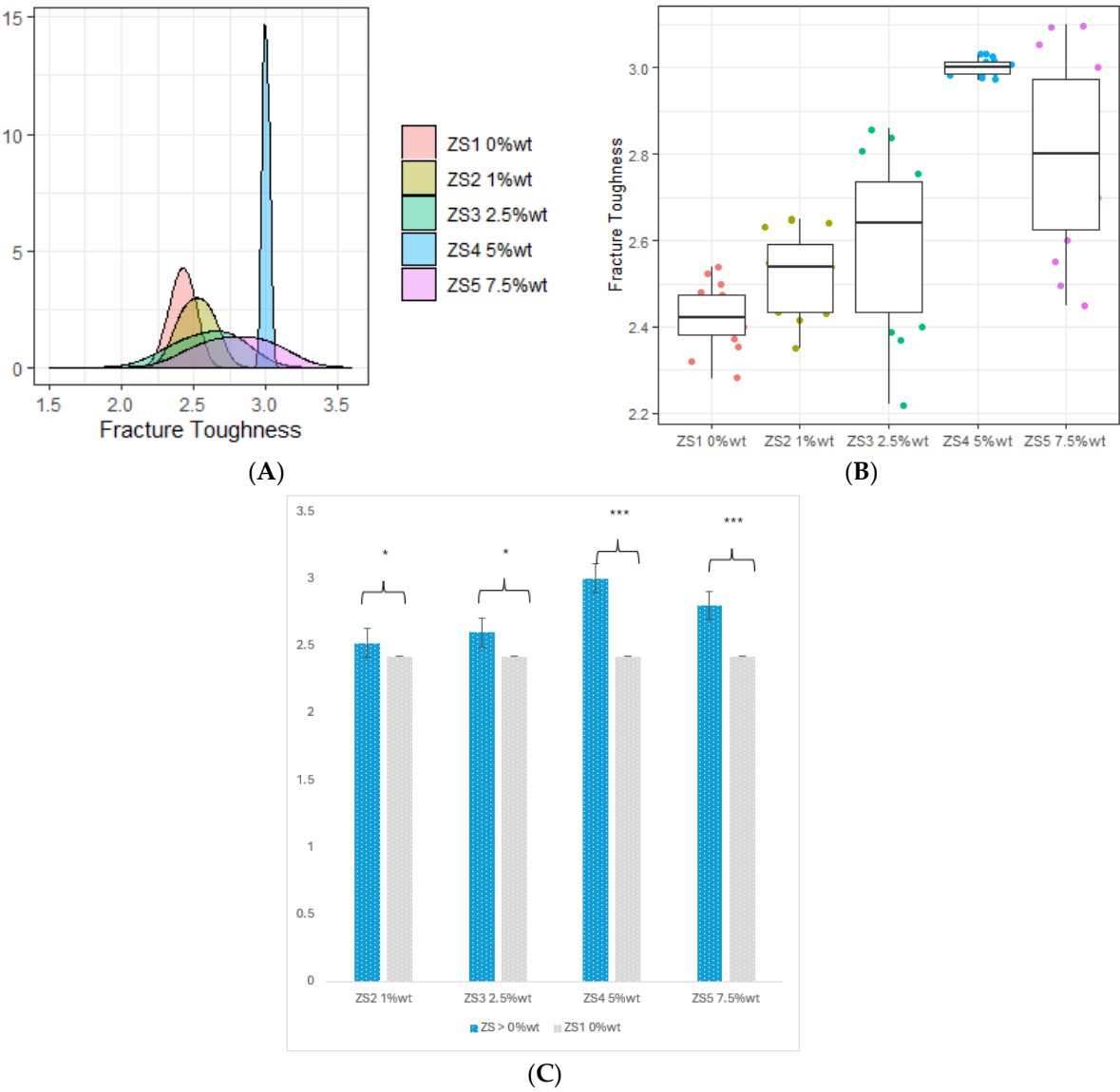

**Figure 9.** (**A**) Distribution of fracture strength in response to different proportions of silver nanoparticle filler; (**B**) box-plot of fracture strength in response to different proportions of silver nanoparticle filler; (**C**) displaying significant differences of $ZrO_2$ 4 %wt. resins and different proportions of silver nanoparticle filler for fracture toughness. * Significant difference; *** highly significant difference.

### 3.4. Vickers Microhardness (VHN)

Upon scrutiny of Table 3, minimal changes in Vickers microhardness (5% and 6%) were detected when incorporating 1 %wt. and 2 %wt. of $ZrO_2$ with the filler, respectively.

An upward trend between the fracture toughness and filler concentration can be seen, as per Figure 10A. In the test comparison using ANOVA, a statistically significant outcome was observed for all additional $ZrO_2$ fillers. As depicted in Figure 10C, the blank resin showed statistically significant differences among the groups of fillers at 1 %wt., 2 %wt., 4 %wt., 8 %wt., and 16 %wt. when compared to the unmodified resin control (0 %wt.) as displayed in Figure 10B.

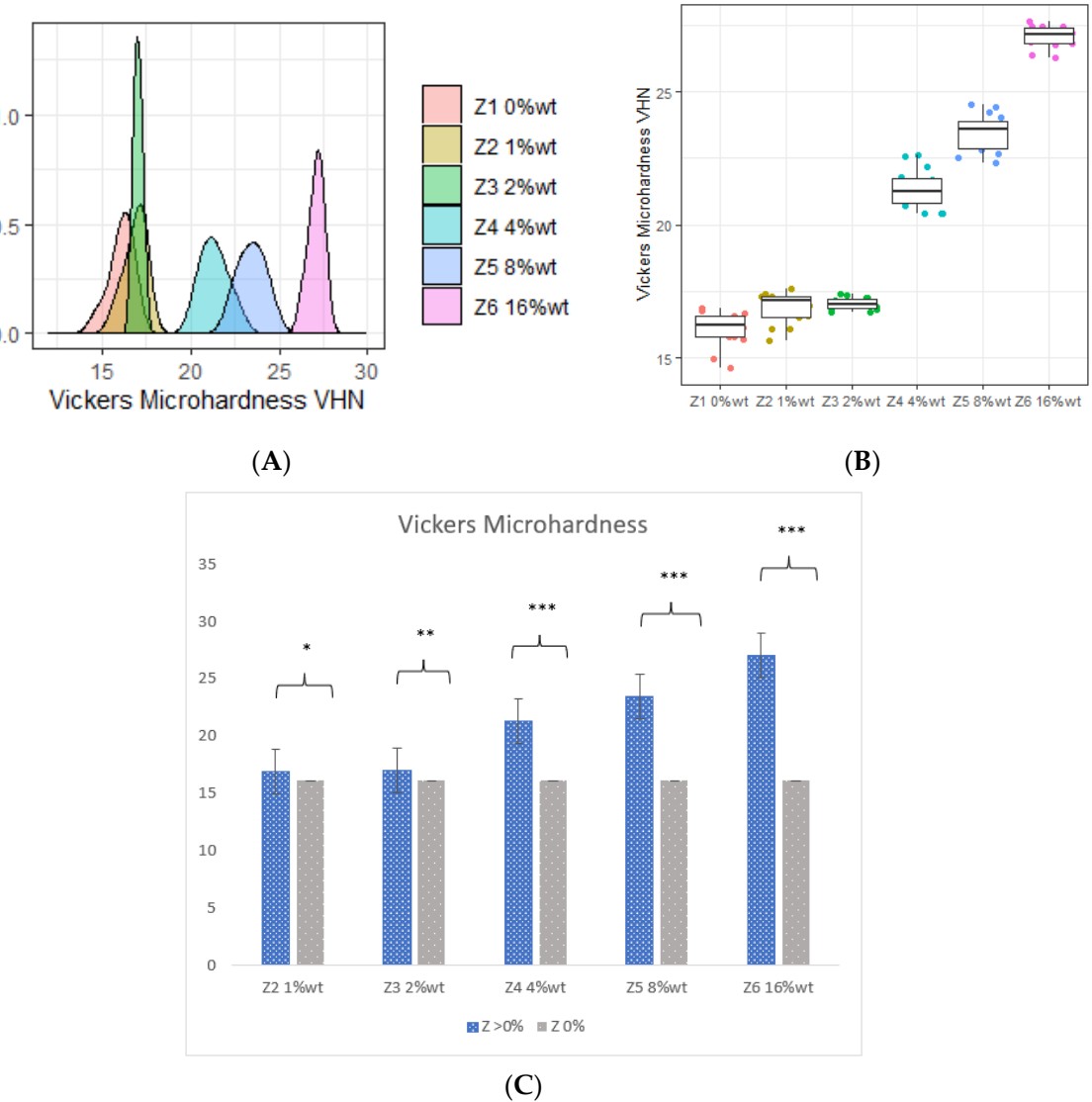

**(A)**

**(B)**

**(C)**

**Figure 10.** (**A**) Distribution of Vickers microhardness in response to different proportions of $ZrO_2$ filler; (**B**) box-plot of Vickers microhardness in response to different proportions of $ZrO_2$ filler; (**C**) displaying significant differences of unmodified resins and different proportions of $ZrO_2$ filler for Vickers microhardness. * Significant difference; ** Mild Significant difference; *** highly significant difference.

Figure 11A captures the filler distribution where the addition of ZS4 at 5 %wt. stands out with the highest VHN, prominently positioned at the peak of the *x*-axis.

Nevertheless, even with the increased ZS5 percentage at 7.5 %wt., a distinct decline of 9.6% in VHN value was observed compared to ZS4 5 %wt., reaching a final measure of 25.511. Figure 11C provides a visual representation of the filler distribution, highlighting ZS4 at 5 %wt. as the standout contributor to VHN, prominently positioned at the zenith of the *x*-axis.

According to the ANOVA test comparison, statistically significant outcomes were observed for all additional ZS fillers. As depicted in Figure 11, the base resin displayed notable statistical distinctions among filler groups at 1 %wt., 2.5 %wt., 5 %wt., and 7.5 %wt. compared to the $ZrO_2$ 4 %wt. resin control (ZS 0 %wt.), as indicated in Figure 11B.

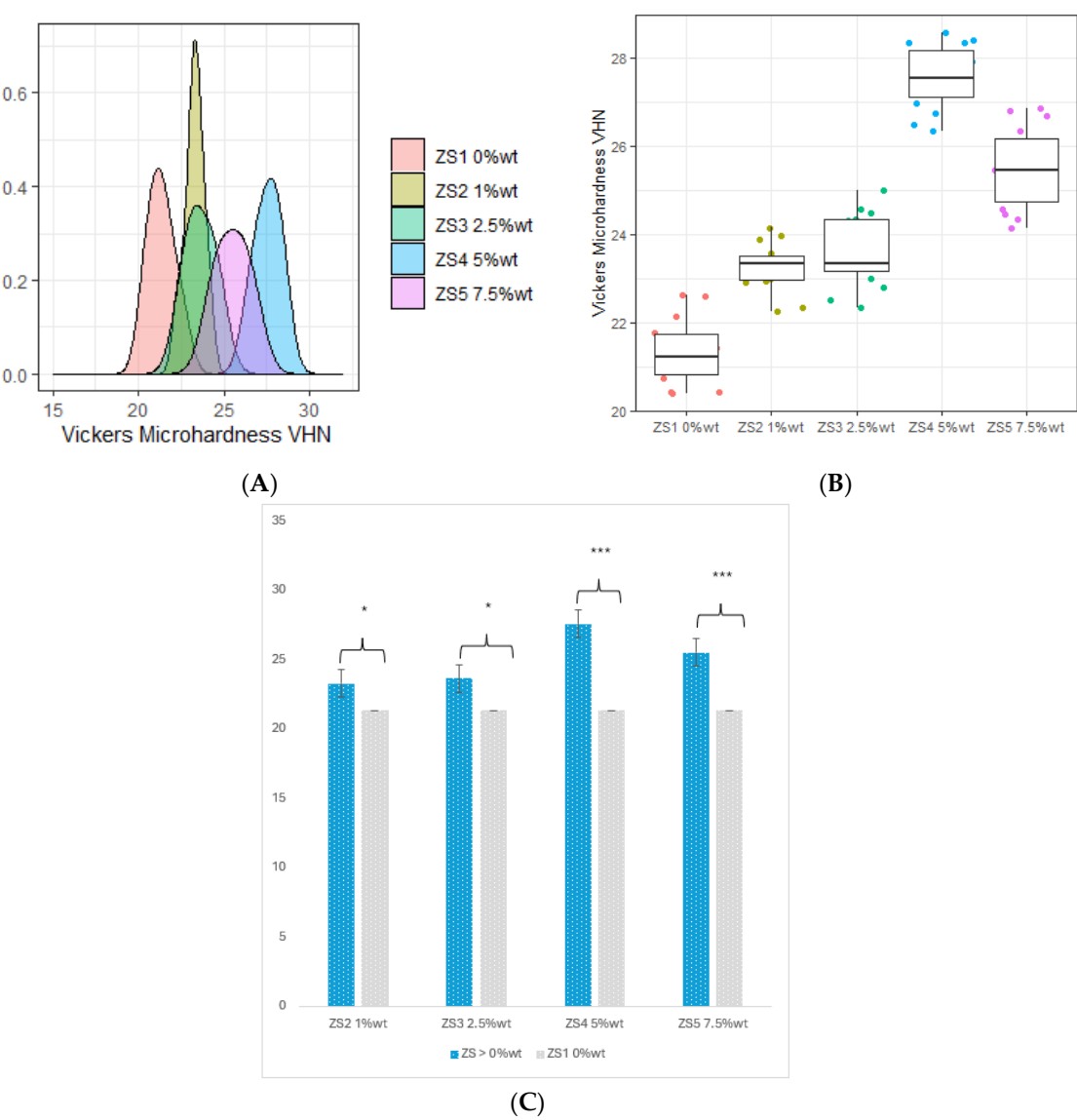

**Figure 11.** (**A**) Distribution of Vickers microhardness in response to different proportions of silver nanoparticle filler; (**B**) Vickers microhardness in response to different proportions of silver nanoparticle filler; (**C**) displaying significant differences of $ZrO_2$ 4 %wt. resins and different proportions of silver nanoparticle filler for Vickers microhardness. * Significant difference; *** highly significant difference.

### 3.5. Microscopical Characterization

By utilizing a scanning electron microscope (SEM) it can be seen that numerous cleavages have emerged inside the pure resin matrix that are free of additives, as illustrated in Figure 12A. Figure 12B shows that $ZrO_2$ nanoparticles display primarily spherical particles with a size below 90 nm. In comparison to the pure resin matrix, the microstructure analysis reveals a significant improvement in the homogeneity of the resin matrix upon the addition of $ZrO_2$ nanoparticles, as shown in Figure 12C. Notably, the addition of $ZrO_2$ nanoparticles to the resin matrix layers significantly decreased cleavages inside the structure, as shown in Figure 12D. Also, Figure 12E shows a large number of $ZrO_2$—HNC/Ag nanoparticle groups intertwined in the resin, which appear as small grains in the image. The homo-

geneity of the structure has improved. Silver nanoparticles are very small in size, and HNC/Ag nanoparticles significantly decrease cleavages inside the structure, regardless of the number of them present. This implies that nanoparticles will disperse in a regulated and desired manner, enhancing structural integrity, and possibly leading to improvements in material qualities.

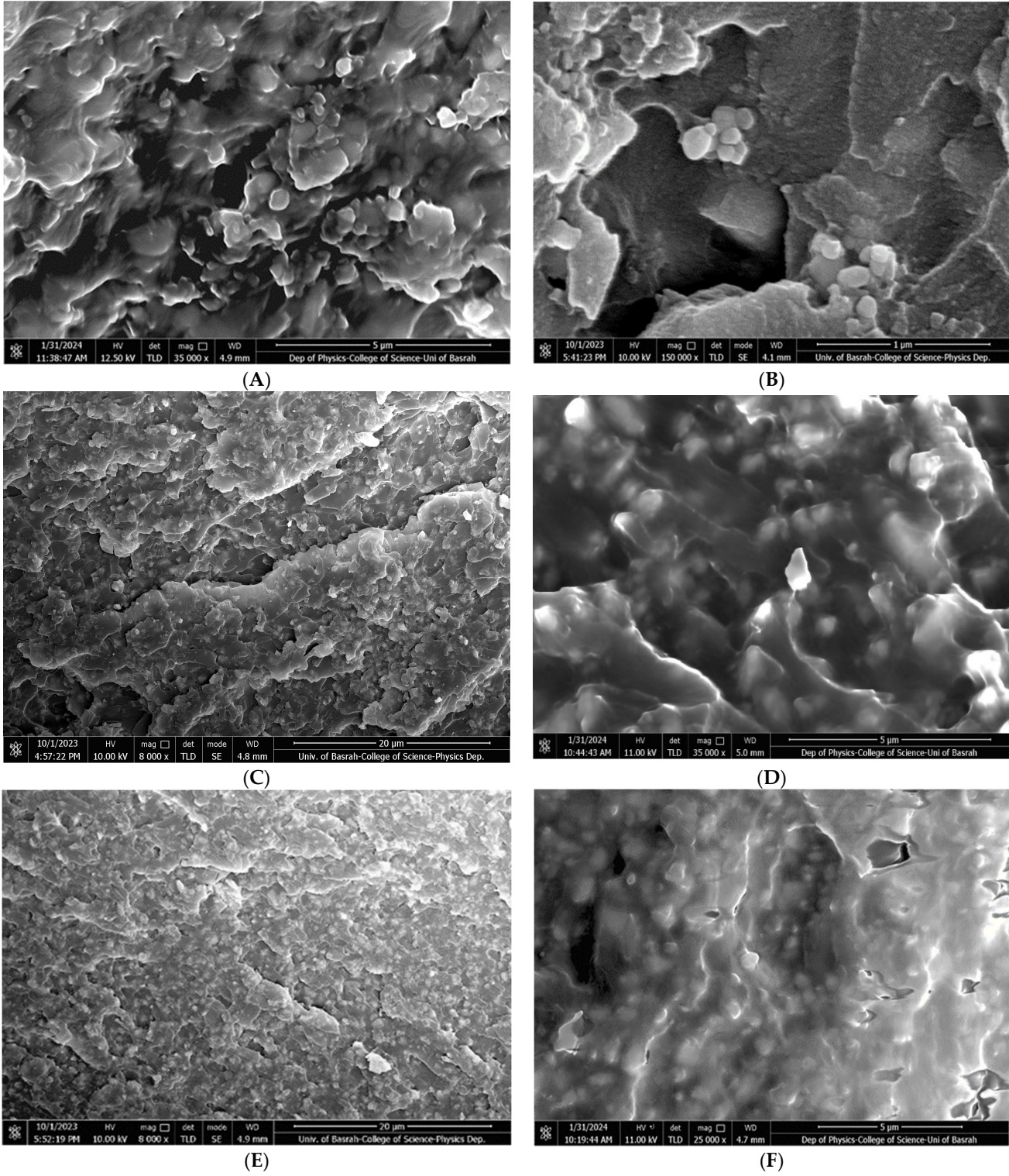

**Figure 12.** SEM microphotographs. (**A**) Resin only; (**B**) $ZrO_2$ nanoparticles; (**C**) homogeneity of $ZrO_2$ in resin matrix; (**D**) $ZrO_2$ covered by resin; (**E**) HNC/Ag distribution in side resin; (**F**) HNC/Ag covered by resin.

## 4. Discussion

This work aims to improve printable liquid resins for permanent dental restorations through the addition of varying concentrations of $ZrO_2$ and HNC/Ag. Additionally, we intend to employ a variety of mechanical and physical assessment techniques to examine the structural, physical, and mechanical properties of the resulting 3D-printable resin. The study's null hypotheses state that adding HNC/Ag and salinized $ZrO_2$ nanoparticles to 3D-printable resin materials in varying amounts will not affect the printed resin-based nanocomposites' flexural characteristics, fracture toughness, and surface microhardness. Researchers have become quite interested in the use of additive technology such as 3D printing in dentistry. The preservation and enhancement of the mechanical and physical characteristics of restorative materials are essential for guaranteeing the restoration's long-term therapeutic success. Studies in this area are therefore very helpful in guiding their application and material improvements. Nanoparticle fillers have become a widely embraced technique in enhancing dental resin composites' mechanical and physical characteristics through strategic utilization [27,28]. Additionally, it has been demonstrated that this method increases flexural strength, tensile strength, fracture toughness, wear resistance, microhardness, and elastic modulus. This method also reduces polymerization shrinkage [29,30]. These properties support the material's overall viability and performance [29,31]. This study was founded by the discovery that incorporating $ZrO_2$/NPs in concentrations higher than 2 %wt. significantly enhanced the flexural strength and flexural modulus of the resin. This finding is consistent with the results of three other investigations which also found that $ZrO_2$/NPs improved resin's flexural properties [22,29,32]. However, Aati et al. verified that incorporating only 1 %wt. and 2 %wt. of $ZrO_2$/NPs was a negligible increase of the flexural strength and modulus of the resin. These findings agree with the present study's results. The highest mean values for flexural strength were observed by 4 %wt., 8 %wt., and 16 %wt. $ZrO_2$/NPs. The outcomes of this test coincide with the studies of Albadr and Rafid as well as Kumar et al., which showed an improvement in FS with increasing concentrations of $ZrO_2$/NPs [33,34].

Compared to the control group (ZS1), all 3D-printed resins containing HNC/Ag NPs had increased FS and FM. The mechanical characteristics were also enhanced as the filler amount was increased from 1% to 5%. Compared to the control group, the 5% HNC/Ag was significantly greater. As the ratio of HNC/Ag climbed to 7.5 %wt., the FS and FM decreased. This could be because of the agglomeration that HNC/Ag created. Increased filler 3D-printing resin causes HNC/Ag to aggregate, which results in a loss of mechanical characteristics. The results of this study indicate that adding nanoparticles to 3D-printed resins increase their FS and FM. This increase is observed when adding zirconia nanoparticles at concentrations of 4% and 8%, as well as HNC/Ag fillers at concentrations of 5%. These findings were reported earlier in [35–37].

The key factor influencing the effectiveness of clinical applications is Vickers microhardness, which is the resistance to permanent deformation and hardness. A high microhardness score makes a material more resistant to scratches and abrasions while also preventing it from deforming easily under different stresses [38]. Improved toughness was observed in the Vickers microhardness (VHN) load capacity of all reinforced specimens compared to the unmodified resin specimen. A distinct upward trend is evident in the relationship between Vickers microhardness and filler concentration, as demonstrated in Table 3.

In this study, it was observed that the hardness was directly proportional to the concentration of NPs incorporated due to their larger contact area with the resin compared to fillers. The effective dispersion of the filler and concentration played a crucial role in achieving this improvement. Notably, specimens with a 4 %wt. filler exhibited significantly higher Vickers microhardness compared to the blank resin, as well as the 1 %wt. and 2 %wt. $ZrO_2$ additions, with a mean value of 17.030 (a 33% increase). However, unlike other properties, Vickers microhardness kept increasing when additional $ZrO_2$ was added to the resin specimens at 8 %wt. and 16 %wt., with mean values of 23.440 and 27.080,

respectively. This created an issue due to the fact that it exceeded the baseline value of this property. A consistent upward trend in the VHN property when applying the dental resin became evident with the introduction of ZS2 at 1 %wt. and ZS3 at 2.5%, as compared to the control resin. The recorded values for these compositions were 21.339, 23.272, and 23.621, respectively, signifying marginal percentage increases of 9.1% and 10.7%. In contrast, the inclusion of ZS4 at 5 %wt. recorded the highest mean value of the VHN test, meaning that this proportion impacts the greatest 27.560, reflecting a substantial 29.2% increase in mean microhardness.

The modified groups of 3D-printing resin with $ZrO_2$/NPs and HNC/Ag NPs showed higher Vickers microhardness. The findings of this investigation are consistent with those of Subhra et al. and Alla et al., who claim that when filler particle concentration increases, surface hardness improves, meaning that filler amount influences material performance. Additionally, these outcomes resemble those of Barot et al., who demonstrated how the size of the silver filler particles affected the internal structure of polymerized PMMA. Similarly, Sokolowski J. et al. observed that an increase in silver nanoparticle concentration led to an increase in Vickers hardness of resin adhesives [39,40]. Additionally, if the right proportions of silver nanoparticles are utilized, they may not have any negative effects on the mechanical properties of composite adhesives that comprise silica nanofillers and silver nanoparticles. Vickers hardness increased as the amount of zirconia filler increased [41], which matched the pattern of an earlier investigation in which PMMA was treated with 0–5% weight percentage zirconium dioxide nanoparticles [42]. Consequently, there may be a direct correlation between surface hardness and zirconia filler concentration [41].

All reinforced specimens had higher load capacities at fracture as compared to the unmodified resin. A $ZrO_2$ addition filler concentration of 4% showed an excellent connection with toughness, which declined gradually between 8% and 16%. Significantly, specimens with 2.5%, 5%, and 7.5% of HNC/Ag addition in the filler had a higher fracture toughness than the control group. Furthermore, Aati et al. discovered up to 3% $ZrO_2$ addition, meaning that there was a positive association between filler concentration and toughness. After that, the toughness dropped to 4% and 5%, respectively. Interestingly, specimens with 3% filler showed a higher fracture toughness than blank resin, with $ZrO_2$ added at 1%, and 2%. Another study by Barot et al. found that there were very few agglomerations and exposed nanotubes in the composite reinforced with 5% HNT/Ag. On the other hand, there was increased irregularity and significant clusters of nanotubes on the cracked surface of 10% HNT/Ag. Agglomerates may function as a structural imperfection that compromises the strength of the resin composite. [43] According to another study, HNT enhances the mechanical properties of resin [44].

Zirconia nanoparticles exhibit robust ionic interatomic bonds with ceramics, acrylics, and restorative resins, resulting in enhanced hardness and strength properties. The inherent characteristics of $ZrO_2$ particles contribute to heightened material hardness through an increase in filler concentration. Jehan et al. highlighted that the incorporation of zirconia nanoparticles has demonstrated an enhancement in mechanical properties, attributed to the improved bonding between the resin matrix and nanoparticles [45].

## 5. Conclusions

The results of this study provide a perspective on the viability of using $ZnO_2$ and HNC/Ag nanoparticles for the enhancement of 3D-printed resin. The physical and mechanical properties of 3D-printed resin were greatly enhanced by different fractional amounts of nanoparticles compared to the unmodified resin. The following conclusions can be drawn:

(A) The incorporation of 4 %wt. $ZrO_2$, 8 %wt. $ZrO_2$, 16 %wt. $ZrO_2$, and 5 %wt. HNC/Ag nanoparticles significantly increased the flexural strength and flexural modulus of the 3D-printed resin, whereas the 7.5 %wt. HNC/Ag decreased the flexural strength and flexural modulus of the resin.

(B) The incorporation of 4 %wt. $ZrO_2$, 2.5 %wt. HNC/Ag, 5 %wt. HNC/Ag, and 7.5 %wt. HNC/Ag nanoparticles significantly increased the fracture toughness of the 3D-printed resin

(C) All fractions of $ZrO_2$ and HNC/Ag nanoparticles significantly increased the micro-hardness of the 3D-printed resin.

**Author Contributions:** Methodology, K.R.D.; Formal analysis, K.R.D.; Resources, K.R.D.; Writing—review & editing, K.R.D. and B.K.A.; Supervision, B.K.A. All authors have read and agreed to the published version of the manuscript.

**Funding:** This research received no external funding.

**Institutional Review Board Statement:** Ministry of Higher Education and Scientific Research, Hawler Medical University, College of Dentistry. Approval number: Den 239.

**Informed Consent Statement:** Not applicable.

**Data Availability Statement:** Data are contained within the article.

**Conflicts of Interest:** The authors declare no conflicts of interest.

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
