# Peer review of "Innovation and Evaluations of 3D Printing Resins Modified with Zirconia Nanoparticles and Silver Nanoparticle-Immobilized Halloysite Nanotubes for Dental Restoration"

_coatings, doi:10.3390/coatings14030310_

Round 1

Reviewer 1 Report

Comments and Suggestions for Authors

Regarding to Manuscript ID: coatings-2898374

Type of manuscript:

Article Title: Innovation and evaluation of 3D printing of resin modified zirconia nanoparticles, silver nanoparticles,immobilized halloysite nanotube for dental restoration  

Authors: Karwan Rashid Darbandi *, Bassam Karem Amin *

The authors prepared some dental composites based on printable CROWNTEC resin, ZrO2 nanaoparticles and Ag nanoparticles immobilized on halloysite nanotubes.

They searched for the optimal composition of the mixtures of resin with fillers, and tested the mechanical properties of the preparations.

Q1: We need to know more about A1 saremco print CROWNTEC product such that to explain the interactions between resin material with ZrO2 and HNC/Ag.

In the Materials section, the origin/supplier, the chemical composition of this resin/class of resinous substances is not specified.

Q2: In section Materials, the following sentence is incomplete:

“For MDP, camphor quinone (Aladdin, China) and 4-dimethylamino-benzoic acid ethyl ester (EDMAB) (Aladdin, China).”

Please revise it.

Also, the explanation of the MDP acronym cannot be found anywhere in the manuscript.

Q3: Synthesis of Silver Nanoparticle section

The meaning of the sentence is not clear:

“This step involving the functionalization of HNC with APTES played a pivotal role in the entire procedure [20, 21].”

Was it about preliminary functionalization of HNC with APTES, followed by immobilization of AgNPs ?

It is strange that it is not specified where halloysite nanotubes HNC comes from, how and who modified it with APTES (references [20; 21] do not reflect any study by the authors of this manuscript).

Q4: It would have been useful or at least interesting if the concentration of AgNPs in HNC had been known, as well as the one released from the HNC as the composite was used.

Q5: Regarding to printer LED light source with a wavelength of 385 nm:

I think the power of the LED source should also be indicated, not just the wavelength.

Does this value coincide with the energy specified in the resin worksheet?

Q6: The explanation of the figure 12 is incomplete:

Figure 12. (A) Resin only; (B) Zro2 nanoparticles; (C)Homogeneity of the Zro2 in resin matrix;(D) Zro2 covered by resin;(E) HNC/Ag distribution in side resin;(F) HNC/Ag cover by resin.”

Maybe: “Figure 12.  SEM microphotographs for (A); ……..(E).

See also small mistake in ZnO2 typing.

Q7: Discussion/ Please revise the sentence:

“The study's null hypotheses state that adding silver nitrate and salinized ZrO2 nanoparticles to 3D-printable resin materials in varying amounts won't affect the printed resin-based nanocomposites' flexural characteristics, fracture toughness, and surface microhardness.”

Silver nitrate was not added to resin. Silver nitrate acted as a precursor for the synthesis of Ag nanoparticles that were later immobilized on HNC.

Author Response

Response to Reviewer 1 Comments

Reviewer 2 Report

Comments and Suggestions for Authors

In this article Darbandi and Amin report 3D printing of resin-modified zirconium nanoparticles, silver nanoparticles, immobilized halloysite nanotubes for dental restorations.

The article reports the procedures for the synthesis and characterization of nanoparticles as well as the results obtained. However, there is no outline of biological tests for the application expressly reported in the title. The authors should carry out some preliminary biological tests to evaluate the dental field application. Only after these preliminary biologicals analyses will it be possible to evaluate the actual innovation made. 

In addition to what reported above, authors are also invited to follow the suggestions below

Introduction

Line 33: remove three-dimensional and put 3D

Lines 34-37: these two sentences say the same. Make it one sentence

Line 43: move DLP after Digital light processing

ZrO2: subscript the 2 (also in the next part of the text)

The purpose of this article is absent. Authors are encouraged to add a sentence or short paragraph at the end of the introduction to explain the purpose of their work

Materials

Check throughout the chapter for the presence of acronyms/acronyms already mentioned or repeated. Leave the full name only when they appear for the first time and then use the acronym

Check subscript in all formulas

For concentrations, indicate only the acronyms and not the extended nomenclatures (e.g. millimolar, always and only indicate with mM since it is well known)

Results

Check subscript in formulas

Line 195: capital letter at the beginning of the sentence

Which ANOVA test did you use and how did you check the normal distribution of the data? It should perhaps be mentioned in the materials section

Author Response

Response to Reviewer 2 Comments

Reviewer 3 Report

Comments and Suggestions for Authors

Comments of coatings-2898374

This study investigates innovation and Evaluations of 3D printing resin modified with zirconia nanoparticles; silver nanoparticles immobilized halloysite nanotube for dental restoration. This is a good work. Before acceptance, some revisions are recommended.

1.     At the end of the introduction, it is necessary to clearly describe the goal and the research methods by which it will be carried out.

2.     Please pay attention to the subscripts of chemical formulas in the text.

3.     Figs. 1~3 should have rulers.

4.     The magnification of SEM measurement should be consistent (Fig. 12).

Author Response

Response to Reviewer 3 Comments

Round 2

Reviewer 2 Report

Comments and Suggestions for Authors

The revised manuscript is well improved. I recommend to publish it.